# Memory-Efficient Backpropagation through Large Linear Layers

## Abstract

In modern neural networks like Transformers, linear layers require significant memory to store activations during backward pass. This study proposes a memory reduction approach to perform backpropagation through linear layers. Since the gradients of linear layers are computed by matrix multiplications, we consider methods for randomized matrix multiplications and demonstrate that they require less memory with a moderate decrease of the test accuracy. Also, we investigate the variance of the gradient estimate induced by the randomized matrix multiplication. We compare this variance with the variance coming from gradient estimation based on the batch of samples. We demonstrate the benefits of the proposed method on the fine-tuning of the pre-trained RoBERTa model on GLUE tasks.

## 1 Introduction

The recent advances in solving NLP tasks are based on the Transformer architecture Vaswani et al. (2017), where the two memory bottlenecks exist in the original formulation. The first one is the attention layer and the second one is the linear layers with large matrices of parameters. The issues of operating with the attention layer in practice are solved with help of a sparsification of the attention matrix Child et al. (2019); Zaheer et al. (2020). A similar challenge in operating with large dense matrices of parameters in linear layers has not been discussed, yet.

Since the propagating of gradient through the linear layer is essentially the computation of matrix by matrix product, we consider the randomization schemes that approximate the target gradient and simultaneously require less memory. There are well-known techniques to compute the approximate matrix multiplication in the literature Drineas et al. (2006). However, typically these techniques are considered from the running time perspective rather than memory consumption. The paper Adelman et al. (2021) proposes to approximate the backward pass through linear layers using randomized matrix multiplication and focuses on the training time and test accuracy of the final model. However, this method has the same memory requirement as the standard one. In the current work, we propose an algorithmic and theoretical justification of a memory-efficient linear layer based on randomized matrix multiplication. The proposed method requires significantly less data to be stored for the computation of the approximate gradient of the loss function with respect to the weight.

We confirm memory reduction and analyze possible convergence deterioration by performing experiments on the finetuning of the pretrained RoBERTa model Liu et al. (2019) on the GLUE tasks Wang et al. (2018). The experimental evaluation of the considered approach demonstrates that the memory reduction does not lead to a significant test accuracy decrease. For some datasets, we have observed that even 90% memory reduction leads to moderate test accuracy decreasing, and sometimes the additional noise is even beneficial for generalization.

The main contributions of this paper are the following.

- Memory-efficient randomized gradient propagation algorithm through large linear layers.
- Theoretical analysis of the gradient variance induced by auxiliary randomized computations.

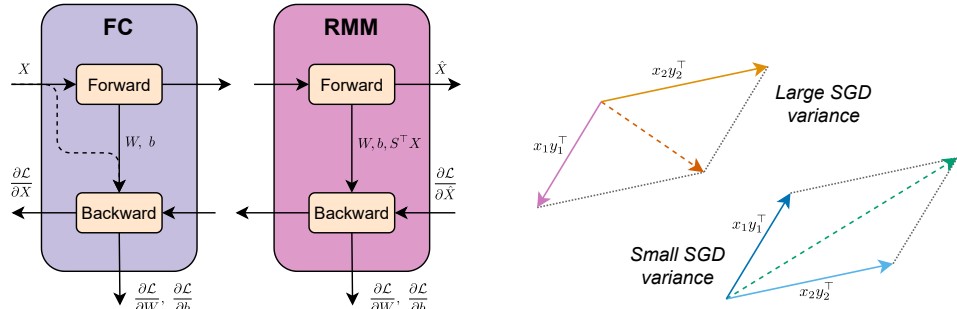

Figure 1: **Left:** Computational graphs for training step in the case of default fully-connected (FC) and randomized linear (RMM) layers. If the standard layer is used we store the whole tensor $X$ for backward (dashed line on the left), while in the proposed randomized version we store only $X_{\text{proj}} = S^\top X$ and a random state (solid line on the right). **Right:** Visual support for Lemma 2.1. If input vectors and output gradients are divergent, the resulting variance estimate of SGD is high. Whenever inputs $X$ and output gradients $Y$ are close, the value of SGD variance is low.

- Empirical analysis of the trade-off between memory efficiency and test accuracy decrease for a number of datasets.
- Experiments are performed in finetuning of pre-trained RoBERTa model on GLUE tasks.

## 2 METHOD

The main building block of neural networks remains a linear layer. It demands a lot of memory and computational resources principally because of multiplication of matrices of considerable sizes. In this section we demonstrate how randomized matrix multiplication alleviates these issues.

First of all, we present our modification to a fully-connected layer. Then we review a common approach of training neural networks and specifically estimation of the stochastic gradient. After that we discuss interplay of different sources of variance and provide some theoretical guarantees. Finally, we give an estimation of memory and arithmetical complexity.

### 2.1 RANDOMIZED BACKWARD PASS FOR A LINEAR LAYER

A linear layer is defined by weights $W \in \mathbb{R}^{N_{\text{out}} \times N_{\text{in}}}$ and biases $b \in \mathbb{R}^{N_{\text{in}}}$. It does nothing but an affine transformation of an input batch $X \in \mathbb{R}^{B \times N_{\text{in}}}$:

$$\hat{X} = XW^\top + \mathbf{1}_B b^\top. \tag{1}$$

Gradients of the loss function with respect to the layer input can be expressed as follows

$$\frac{\partial \mathcal{L}}{\partial X} = \frac{\partial \mathcal{L}}{\partial \hat{X}} W, \tag{2}$$

and gradients of the loss function with respect to layer weights are

$$\frac{\partial \mathcal{L}}{\partial W} = \left(\frac{\partial \mathcal{L}}{\partial \hat{X}}\right)^\top X, \quad \frac{\partial \mathcal{L}}{\partial b} = \left(\frac{\partial \mathcal{L}}{\partial \hat{X}}\right)^\top \mathbf{1}_B. \tag{3}$$

**Analysis of memory consumption.** In standard implementation the input tensor $X$ is *stored entirely* until the gradient over $W$ is calculated. As in Adelman et al. (2021) we suggest to replace the matrix multiplication in equation 3 with its randomly sampled counterpart, but with a key difference: **our goal is not to speedup the computation, but to save**

---

**Algorithm 1** Forward and backward pass through a linear layer with a randomized matrix multiplication.

---

**function** FORWARD($X$, $W$, $b$)
    $\hat{X} \leftarrow XW^\top + \mathbf{1}_B b^\top$
    Generate pseudo random number generator (PRNG) state and random matrix $S$
    $X_{\text{proj}} \leftarrow S^\top X$
    Save $X_{\text{proj}}$ and PRNG state for the backward pass.
    **return** $Y$
**end function**

**function** BACKWARD($\partial_{\hat{X}} \mathcal{L}$, $W$, $b$, $X_{\text{proj}}$)
    $\partial_X \mathcal{L} \leftarrow \partial_{\hat{X}} \mathcal{L} \cdot W^\top$
    Rematerialize matrix $S$ from the PRNG state saved in the forward pass.
    $\partial_W \mathcal{L} \leftarrow \left( \partial_{\hat{X}} \mathcal{L}^\top \cdot S \right) \cdot X_{\text{proj}}$
    $\partial_b \mathcal{L} \leftarrow \partial_{\hat{X}} \mathcal{L}^\top \mathbf{1}_B$
    **return** $\partial_X \mathcal{L}$, $\partial_W \mathcal{L}$, $\partial_b \mathcal{L}$
**end function**

---

**the memory during the training stage**. Namely (see, i.e., Drineas et al. (2006)) we have

$$\frac{\partial \mathcal{L}}{\partial W} = \mathbb{E}_S \left[ \left( \frac{\partial \mathcal{L}}{\partial \hat{X}} \right)^\top S S^\top X \right] = \mathbb{E}_S \left[ \left( \frac{\partial \mathcal{L}}{\partial \hat{X}} \right)^\top S X_{\text{proj}} \right], \tag{4}$$

where $X_{\text{proj}} = S^\top X \in \mathbb{R}^{B_{\text{proj}} \times N_{\text{in}}}$ is calculated during the forward pass and stored instead of $X$ (see Algorithm 1). In order for this to be possible, matrix $S$ *has to be independent from* $Y = \dfrac{\partial \mathcal{L}}{\partial \hat{X}}$. In Adelman et al. (2021) the construction of $S$ requires the knowledge of the norms of the rows of $Y$, so we can not precompute $XS$.

The only requirement for the random matrix $S \in \mathbb{R}^{B \times B_{\text{proj}}}$ is that it has to satisfy

$$\mathbb{E} \, SS^\top = I_{B \times B},$$

where $I_{B \times B}$ is $B \times B$ identity matrix. Note, although $S$ is needed in the backward pass (it should be the same as in the forward pass), it is not stored explicitly but rematerialized from the stored random seed. We will refer to the approximate matrix multiplication algorithm used in equation 4 as **Randomized Matrix Multiplication (RMM)**.

Different random distributions can be used to generate matrix $S$. In this work, we consider a Gaussian random matrix,

$$S = \frac{1}{\sqrt{B_{\text{proj}}}} P, \tag{5}$$

where the elements of $P$ are i.i.d Gaussian random variables with zero mean and unit variance. We also tested other variants such as Subsampled Orthonormal with Random Signs (SORS) matrices Iwen et al. (2021). They come with fast matrix-by-vector product but the accuracy drop is higher, so we leave this for future studies and do not report it here.

## 2.2 STOCHASTIC GRADIENT ESTIMATION

We have a randomized computation of the gradient; how accurate this should be? In standard tasks, the approximation should approximate the target really accurate, i.e. with high relative accuracy. Randomized matrix multiplication error decays like $\mathcal{O}(B_{\text{proj}}^{-0.5})$ (the exact estimates will be described in Section 2.3), so it may seem it is not a good idea. However, in the framework of stochastic gradient descent (SGD) we already have a noisy estimation of the gradient which is induced by sampling of the dataset, i.e. this approximation has some variance. Thus, it is natural to require that the variance induced by the randomized approximation is of the same order, as the variance induced by the stochastic estimation of the gradient. Moreover, higher total variance of the gradient estimate does not necessary

mean that the convergence of the overall optimization may be worse, since the noise can be beneficial. In order to estimate the effect of RMM on the variance, we need to have a certain estimate of the variance of the gradient estimation.

Suppose, we have the following optimization problem in a form of finite sample average minimization:

$$f(\boldsymbol{\theta}) = \frac{1}{N} \sum_{i=1}^{N} f_i(\boldsymbol{\theta}) \to \min_{\boldsymbol{\theta} \in \mathbb{R}^p}. \tag{6}$$

Usual approach to deal with such problem involves stochastic first order methods, where instead of full gradient computation $\nabla f(\boldsymbol{\theta}) = \frac{1}{N} \sum_{i=1}^{N} \nabla f_i(\boldsymbol{\theta})$ one can use stochastic approximation of this vector

$$g(\boldsymbol{\theta}) = \frac{1}{n} \sum_{j=1}^{n} \nabla f_{i_j}(\boldsymbol{\theta}) \to \min_{\boldsymbol{\theta} \in \mathbb{R}^p}, \tag{7}$$

where $\mathcal{I} = \{i_1, \ldots, i_j, \ldots, i_n\}$ is sampled uniformly from original set of indices $\{1, \ldots, N\}$. The number $n$ is the batch size. For convenience, we can deal with this randomness considering a stochastic gradient vector $g(\boldsymbol{\theta}) = g_\xi$ as follows.

$$g_\xi = \frac{1}{n} \sum_{i=1}^{N} \nabla f_i(\boldsymbol{\theta}) \xi_i, \quad \text{where } \xi_i = \begin{cases} 1, & \text{if } i \in \mathcal{I} \\ 0, & \text{otherwise.} \end{cases} \tag{8}$$

The estimate in equation 8 can be viewed as an empirical mean of the vector random variable. Thus, we can also build an empirical estimator of the variance of this random variable, and use it as a guidance for the variance of the RMM model. We will do it specifically for the linear layer, since in this case very simple and intuitive formulas can be obtained.

### 2.3 Variance of Stochastic Gradient Estimate

With background given in Section 2.2 we are able to discuss our main theoretical contribution. One can follow detailed derivations in Appendix A. The first observation that we make is that the exact gradient computed for a given batch can be viewed as an empirical mean estimate of a random variable, i.e. it has a certain amount of noise. The randomized approximation introduces additional noise into the picture, which can be either smaller, than the noise from the finite sample size (in this case we expect the convergence to stay the same) or larger. In the latter case, the effect of the additional noise can sometimes play the role of regularizer. Theory of SGD convergence and generalization is rapidly developing, see for example Keskar et al. (2019); Jastrzebski et al. (2017); Hoffer et al. (2017); Cheng et al. (2020); Li et al. (2021). In some settings, generalization can be even improved by injecting additional noise Hoffer et al. (2017); Cheng et al. (2020); Li et al. (2021).

The benefits of the noise in SGD are quite well understood, however we are not aware of any practical estimators of this noise. The following Lemma shows how it can be done using a very standard statistical estimator of the variance.

**Lemma 2.1** (Aposteriori variance of SGD)**.** *Let $X \in \mathbb{R}^{B \times N}$ and $Y \in \mathbb{R}^{B \times M}$ be the input to the linear layer in the forward pass and the input to it in the backward pass (B here is the batch size). Then, we can estimate the variance of the noise induced by a random selection of the samples as*

$$D_{\text{SGD}}^2(X, Y) = \frac{B}{B-1} \sum_{k=1}^{B} \|x_k\|^2 \|y_k\|^2 - \frac{\|X^\top Y\|_F^2}{B-1}, \tag{9}$$

*where $x_k = X^\top e_k$, $y_k = Y^\top e_k$, $k = 1, \ldots, B$, i.e., $x_k$ and $y_k$ are the columns of $X^\top$ and $Y^\top$, respectively.*

The meaning of the estimate equation 9 is very simple. In the first term we have the norms of the per-example gradients, and the last term is the scaled norm of the gradient for the entire batch. If the latter is small, but the norms of per-example gradients are large, then we have high variance of the SGD (see Section 1). Intuitively, the Lemma 2.1 can be viewed as a generalization of a sample variance in the stochastic gradient estimation (for full derivation see Appendix A.1).

**Lemma 2.2** (Apriori variance of RMM). *Let $X \in \mathbb{R}^{B \times N}$ and $Y \in \mathbb{R}^{B \times M}$, then the variance of a randomized matrix multiplication through a matrix $S \in \mathbb{R}^{B \times B_{\mathrm{proj}}}$ with i.i.d. elements following the normal distribution $\mathcal{N}(0, B_{\mathrm{proj}}^{-0.5})$ defined as*

$$D^2(X, Y) = \mathbb{E}_S \, \|X^\top S S^\top Y - X^\top Y\|_F^2 \tag{10}$$

*can be evaluated as follows*

$$D_{\mathrm{RMM}}^2(X, Y) = \frac{\|X\|_F^2 \|Y\|_F^2 - \|X^\top Y\|_F^2}{B_{\mathrm{proj}}}. \tag{11}$$

The proof can be found in the Appendix A.2.

**Theorem 2.3** (Upper bound of variance). *In the conditions of Lemma 2.1 and Lemma 2.2 the in-sample variance $D_{\mathrm{SGD}}$ and the variance $D_{\mathrm{RMM}}$ induced by a randomized subsampling are tied with the following inequality*

$$\frac{B_{\mathrm{proj}}}{B - 1} \frac{D_{\mathrm{RMM}}^2(X, Y)}{D_{\mathrm{SGD}}^2(X, Y)} \leq \frac{\alpha + 1}{\alpha}, \tag{12}$$

*where*

$$\alpha = \frac{\|X^\top Y\|_F^2}{\|X\|_F^2 \|Y\|_F^2} \in [0, 1]. \tag{13}$$

The proof can be found in the Appendix A.3. It is worth noting, that the parameter $\alpha$ can actually be zero in the case $X^\top Y = 0$ leading to a non-bounded variation. Let assume the following simple example with $B = 2$:

$$X = \begin{bmatrix} 1 & 0 \\ -\varepsilon & 0 \end{bmatrix}, \quad Y = \begin{bmatrix} 1 & 0 \\ \varepsilon^{-1} & 0 \end{bmatrix}, \quad X^\top Y = 0, \tag{14}$$

with some parameter $\varepsilon > 0$. So, the estimated variations are:

$$(B - 1)D_{\mathrm{SGD}}^2(X, Y) = 4, \tag{15}$$

and

$$B_{\mathrm{proj}} \, D_{\mathrm{RMM}}^2(X, Y) = 2 + \varepsilon^2 + \varepsilon^{-2}. \tag{16}$$

Therefore, their ratio can be any arbitrary large number, and "sample" variance of SGD can be much smaller than the one introduced by the RMM. In practice, however, we did not observe such cases. A natural explanation is that for the minima of the loss function that generalizes well the norm of $Y$ will be also small (the norm of $X$ can be made bounded by, i.e., batch normalization) since the gradient with respect to almost every sample will be small.

## 2.4 Memory Footprint and Arithmetic Complexity

**Computational Complexity**   General matrix multiplication (matmul) $AB$ takes $O(nml)$ floating-point operations for $A \in \mathbb{R}^{n \times m}$ and $B \in \mathbb{R}^{m \times l}$. No extra space is required except for storing resulting matrix. Costs estimations are summarized in Table 1.

**Memory Requirements**   Default implementation of a fully-connected layer stores input tensor $X$, which is used both for the forward and the backward passes. It requires $O(BN_{\mathrm{in}})$ extra memory in addition to a weight tensor. Our modification of a fully-connected layer stores a compressed input tensor instead which requires $O(B_{\mathrm{proj}}N_{in})$ memory. Please note, that random matrices are rematerialized when needed from a certain pseudorandom number generator, i.e., the random seed with $O(1)$ memory consumption. In other words our approach reduces memory footprint by $\rho^{-1} = B/B_{\mathrm{proj}} \geq 1$ times for input tensors of all the linear layers.

Table 1: Summary table for Section 2.4 and Appendix C.1.1. Comparison of memory usage required to store input activations between baseline and randomized FC-layers. Columns FORWARD and BACKWARD show how the costs of the compared approaches are split between forward and backward passes.

| | MEMORY | FORWARD | BACKWARD |
|---|---|---|---|
| No RMM | $BN_{\text{in}}$ | 1 | $BN_{\text{in}}N_{\text{out}}$ |
| RMM | $B_{\text{proj}}N_{\text{in}}$ | $BB_{\text{proj}}N_{\text{in}}$ | $B_{\text{proj}}N_{\text{out}}(B+N_{\text{in}})$ |

Table 2: Performance in fine-tuning on GLUE benchmark for different compression rates $\rho$ (number of dimensions of projection space) in percents. The top row (No RMM) corresponds to a baseline implementation without compression.

| | COLA | MNLI | MRPC | QNLI | QQP | RTE | SST2 | STSB | AVG |
|---|---|---|---|---|---|---|---|---|---|
| No RMM | 60.51 | 87.56 | 89.30 | 92.60 | 91.69 | 78.52 | 94.09 | 90.37 | 85.58 |
| 90% | 59.75 | 87.58 | 88.64 | 92.75 | 91.47 | 77.50 | 94.72 | 90.39 | 85.35 |
| 50% | 59.45 | 87.58 | 88.73 | 92.56 | 91.41 | 77.18 | 94.61 | 90.32 | 85.23 |
| 20% | 57.46 | 87.59 | 87.99 | 92.62 | 91.16 | 76.26 | 94.43 | 90.06 | 84.70 |
| 10% | 57.53 | 87.51 | 88.30 | 92.55 | 90.93 | 75.45 | 94.27 | 89.90 | 84.56 |

## 3 EXPERIMENTS

In this section, we evaluate the performance of the proposed modification of linear layers by comparing it with default implementation. All randomized matrix multiplications are implemented with PyTorch (Paszke et al., 2019) in Python (see supplementary materials for reference implementation). We use pretrained RoBERTa-base model from HuggingFace's Transformers Wolf et al. (2020). Model fine-tuning on GLUE tasks is conducted in a single GPU setting with NVIDIA Tesla V100 SXM2 16 GB. We use the same training setting and model hyperparameters for RoBERTa model which are in Fairseq Ott et al. (2019).

Despite that our primal interest lies in the area of Transformer-based models in NLP domain, we carried out some auxiliary experiment in order to demonstrate universality of RMM and its applicability to different domains and diffrent tasks (see Appendix B).

We rewrite implementation of fully-connected layer in PyTorch with modification to forward pass and backward pass caching compressed input $S^\top X$ and PRNG state $\mathcal{G}$ between passes. Our implementation allows to control compression rate $\rho$ (dimension of random projection proportional to batch size) or to fix a number of dimensions $B_{\text{proj}}$. In both regimes we are able to clamp $B_{\text{proj}}$ in some desired interval. For a sake of clarity, we stick to specifying $\rho$ instead of fixing exact value of $B_{\text{proj}}$ in order to compress uniformly across all layers in a model.

### 3.1 PERFORMANCE ON GLUE BENCHMARK

In these experiments we measure performance degradation in fine-tuning of base RoBERTa model on GLUE benchmark depending on compression rate $\rho$ (see Table 2). Randomized dense layer demonstrates moderate degradation of evaluation metrics. Compression in 5–10 times results in insignificant drop of performance for almost all GLUE tasks.

### 3.2 MEMORY EFFICIENCY

Although fully-connected layer is a common for Transformer architecture and it holds a major share of total memory usage in training time, there is other solid memory consumers. So, we measure actual memory footprint reduction in relation to compression rate $\rho$ (see Table 3). In this experiment setting we train RoBERTa on GLUE tasks with varying compression rate $\rho$ and batch size $B$. Important observation is that compression in 5–10 times cuts overall runtime memory by 10–20%.

Table 3: **Left:** Maximal memory usage during training on GLUE tasks and memory economy for different compression rates $\rho$ and a baseline implementation (No RMM). **Right:** Comparison of different randomized matmul variants. All alternatives are trained on CoLA task. Lower compression rate $\rho$ means lower memory usage.

| Task | Batch | Rate | Mem, GiB | Save, % | MatMul | Rate | Score | Time |
|------|-------|------|----------|---------|--------|------|-------|------|
| MRPC | 128 | — | 11.3 | 0.0 | No RMM | — | 60.90 | 08:44 |
| | | 50% | 10.6 | 6.3 | DCT | 50% | 59.17 | 16:26 |
| | | 20% | 9.2 | 19.3 | | 20% | 58.81 | 16:37 |
| | | 10% | 8.7 | 23.3 | | 10% | 53.38 | 17:24 |
| QNLI | 16 | — | 11.7 | 0.0 | DFT | 50% | 59.05 | 12:20 |
| | | 50% | 11.2 | 4.2 | | 20% | 60.60 | 11:42 |
| | | 20% | 10.4 | 11.6 | | 10% | 47.62 | 12:25 |
| | | 10% | 10.1 | 13.8 | Gauss | 50% | 58.60 | 10:36 |
| SST2 | 256 | — | 13.3 | 0.0 | | 20% | 57.79 | 10:02 |
| | | 50% | 12.5 | 6.1 | | 10% | 56.52 | 10:03 |
| | | 20% | 10.5 | 20.8 | Radem. | 50% | 62.38 | 15:27 |
| | | 10% | 9.9 | 25.5 | | 20% | 59.11 | 15:38 |
| | | | | | | 10% | 55.50 | 15:43 |

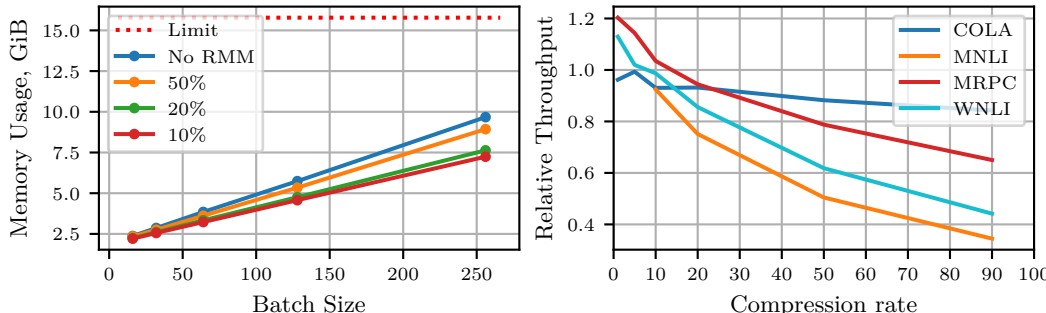

Figure 2: Left: Peak memory usage depending on batch size during training of RoBERTa model for one epoch on CoLA task. Right: Relative throughput of randomized FC layers depending on the compression rate in training time (throughput is a number of samples per second). Relative throughput value above 1 means that a model shows better performance than reference model without randomization.

Also, we carry out experiments to validate memory usage in our implementation with varying of batch size $B$. According to Section 2.4 we save only $O(B_{\mathrm{proj}} N_{\mathrm{in}})$ memory for the backward pass. So, near-linear scaling of memory usage for different compression rates $\rho$ as batch size growth confirms correctness of the implementation (see Figure 2).

### 3.3 Empirical Variance Estimation

In this section we explore empirically variance estimation behaviour (see Section 2.3). We use our common experimental settings where linear layers with randomized backward pass were used. We pick a fully-connected layer and estimate variations equation 9 and equation 11 during training (see Figure 3).

The behaviour of the variance estimators is interesting on its own: the variance slowly increases with the number of steps, whereas as we have seen, the norm of the gradient ($X^\top Y$ term) is very small. This means, that the whole dynamics is governed by the noise terms, i.e. the parameters undergo a diffusion process. The relative behaviour of $D_{\mathrm{SGD}}^2$ and $D_{\mathrm{RMM}}^2$ is also similar and converges to a certain constant. For other layers the picture is very similar. One can find additional experiments in Appendix B.1.

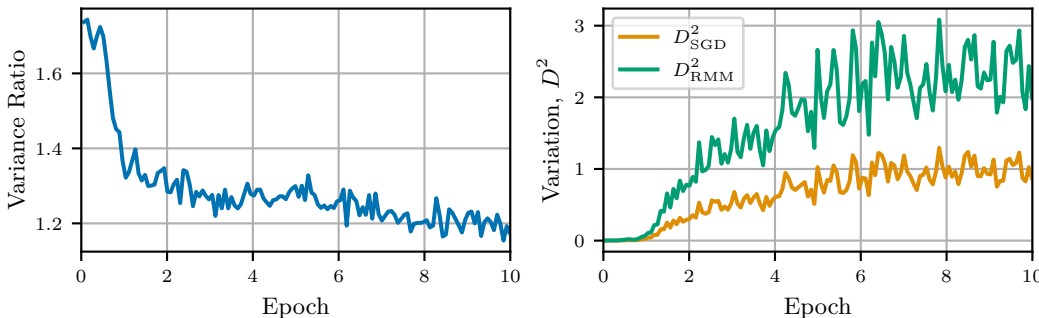

Figure 3: Evolution of variance ratio from the left-hand side of inequality equation 12 (top) and variances estimates equation 9 and equation 11 (bottom) during fine-tuning on CoLA for batch size $B = 64$ and compression rate $\rho = 0.5$.

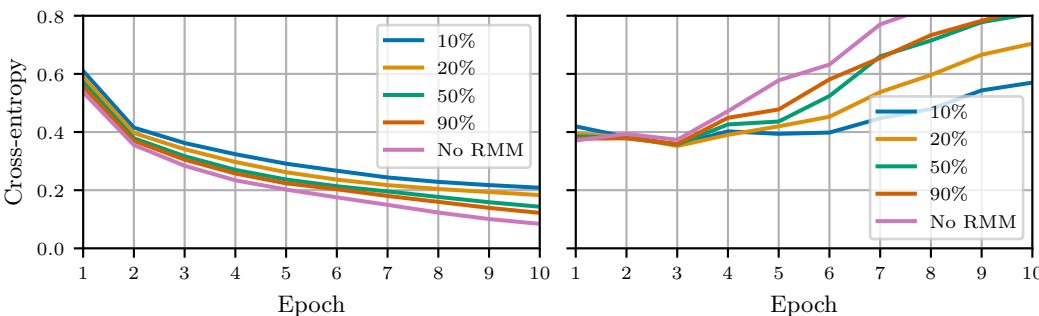

Figure 4: Fine-tuning RoBERTa on MNLI task from GLUE. Cross-entropy loss on training set (left) and evaluation set (right).

### 3.4 Learning Curves

In this subsection we empirically study influence of randomized fully-connected layer on training. Namely, we discover behaviour of cross-entropy loss on training set and evaluation set depending on compression rate $\rho$. We found that loss curve changes smoothly as compression rate declines (see Figure 4). Decreasing of compression rate results in increasing training loss and flattening evaluation loss. However, overfitting point is nearly the same.

### 3.5 Comparison of Randomized MatMuls

In order to reduce computation cost we examine a variety of randomized matrix multiplication implementations. Among matmul implementations we considered, there are sampling of random matrix $S$ from either Gaussian or Rademacher distribution and applying discrete Fourier Transform (DFT) or Discrete Cosine Transform (DCT). In comparison to other approaches, DCT and DFT have theoretically computational advantage because of their regular structure. DFT- and DCT-based matmuls allow to perform multiplication by random matrix $S$ in $O(BN \log B)$ operations instead of $O(B^2 N)$. All alternatives requires the same memory space.

In the case of Gaussian randomized matmul we sample i.i.d. elements of matrix $S$ from normal distribution $\mathcal{N}(0, B_{\text{proj}}^{-0.5})$. The same remains true for the instance of Rademacher distribution which has the following probability mass function $\mathbb{P}(n) = \dfrac{1}{2}$, $n = \pm 1$. The only difference is that we should force unbiasedness condition $\mathbb{E}\, SS^T = I$ with proper normalization.

We found that different matmul variants demonstrate consistent performance degradation of a moderate level as compression rates $\rho$ decreases (see Table 3). Nevertheless, varying

training time across alternatives indicates that naive high-level implementation in PyTorch is not good enough and low-level optimizations are needed.

## 3.6 Computational Time

In order to make experimental study as comprehensive as possible, we investigate computational efficiency empirically although it is not our primary target. We use the standard experimental settings and measure a number of samples processed per second (throughput) in training time (see Figure 2). As it was mentioned in Appendix C.1.1 randomization of linear layer has worse computational complexity in terms of batch size $B$. However, there is small enough compression rate $\rho$ such that randomized dense layer becomes computationally efficient. Moreover, we empirically found that our randomization is faster if $\rho \le 0.1$.

## 4 Related Works

A close work to ours is Adelman et al. (2021), where another variant of randomized multiplication is used to speedup the backward pass. Our goal is to reduce memory and we also provide theoretical analysis of the variance, which sheds light on effect of approximate gradient computations. In Oktay et al. (2020) the concept of randomized automatic differentiation has been proposed.

Randomized matrix multiplication has a long history, which goes back to Freivalds (1977) where probabilistic verification of matrix multiplication has been proposed. In Drineas et al. (2006) the score-based randomized algorithm has been proposed and analyzed. Improved algorithms for matrix multiplication have been proposed in Boutsidis & Gittens (2013), where different fast algorithms have been studied for the sampling matrix based on the results of Tropp (2011) for subsampled orthogonal transforms.

Another line of research focuses on other algorithms for approximation of matrix multiplications, to name a few relevant papers Pagh (2013) where the hashing functions have been used and in Blalock & Guttag (2021) hashing functions are learned from the data. Excellent review for the probabilistic linear algebra can be found in Martinsson & Tropp (2020).

## 5 Conclusion and Future Work

We propose a drop-in replacement for a linear layer in deep neural network with randomized backward operation that reduces the amount of memory, required to be stored during backpropagation. The algorithm is based on a randomized matrix multiplication. We provide theoretical bounds on the additional variance introduced by randomization compared to the inherent noise in the SGD, provide bounds on this noise and computable estimates. In fine-tuning of a Transformer-based model on different GLUE tasks we show that we get reduction in the peak memory while maintaining the accuracy of the model.

There are several directions we would like to study in future work. First of all, we would like to get stable and robust implementations of randomized matrix multiplication with matrices $S$ that allow fast matrix-by-vector product. The Subsampled Orthogonal with Random Signs seems to be a good option, but the variance of such estimators in our experiments was quite large, so the $B_{\mathrm{proj}}$ has to be selected larger. Thus, we would like to study other options. For Transformer-based models the number of rows in $X$ is actually the product of the batch size and sequence length, i.e. it is quite large; however, we do not use this structure. One option is to use tensor-product sampling matrices to reduce complexity of multiplication by $S$.

Second direction is the deeper study of the variance estimator and its connection to the learning rate schedule. Given a good estimate of the variance, one can try to theoretically justify a specific learning rate schedule to maintain a certain level of noise in training.

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
