## A    PROOFS

### A.1    PROOF OF THE LEMMA 2.1

**Lemma 2.1** (Aposteriori variance of SGD). *Let $X \in \mathbb{R}^{B \times N}$ and $Y \in \mathbb{R}^{B \times M}$ be the input to the linear layer in the forward pass and the input to it in the backward pass (B here is the batch size). Then, we can estimate the variance of the noise induced by a random selection of the samples as*

$$D_{\text{SGD}}^2(X, Y) = \frac{B}{B-1} \sum_{k=1}^{B} \|x_k\|^2 \|y_k\|^2 - \frac{\|X^\top Y\|_F^2}{B-1}, \tag{9}$$

*where $x_k = X^\top e_k$, $y_k = Y^\top e_k$, $k = 1, \ldots, B$, i.e., $x_k$ and $y_k$ are the columns of $X^\top$ and $Y^\top$, respectively.*

*Proof.* Unbiased estimate for the stochastic gradient is

$$\overline{\frac{\partial \mathcal{L}}{\partial w}} = \frac{1}{B} \sum_{k=1}^{B} B x_k y_k^\top, \tag{17}$$

which can be seen as an empirical mean of a matrix random variable

$$Z = B x y^\top, \tag{18}$$

with the following average value

$$\overline{Z} = X^\top Y. \tag{19}$$

In order to estimate the variance of the random variable $Z$, we use the empirical variance estimator

$$D_Z^2(X, Y) = \overline{\|Z - \overline{Z}\|_F^2} \approx \mathbb{E}\|Z - EZ\|_F^2. \tag{20}$$

The variance of the empirical mean is connected to it as

$$D_{\text{SGD}}^2(X, Y) = \frac{1}{B-1} D_Z^2(X, Y). \tag{21}$$

The unbiased estimator of the variance is then evaluated as

$$D_{\text{SGD}}^2(X, Y) = \frac{1}{B(B-1)} \sum_{k=1}^{B} \left\| B x_k y_k^\top - \overline{Z} \right\|_F^2. $$

The square of Frobenius norm can be rewritten with a subsequent summation over $k$

$$D_{\text{SGD}}^2(X, Y) = \frac{B}{B-1} \sum_{k=1}^{B} \|x_k y_k^\top\|_F^2 + \frac{1}{B-1} \|\overline{Z}\|_F^2 - \frac{2}{B-1} \left\langle \sum_{k=1}^{B} x_k y_k^\top, \overline{Z} \right\rangle_F, \tag{22}$$

where $\langle \cdot, \cdot \rangle_F$ is the Frobenius scalar product and

$$\left\langle \sum_{k=1}^{B} x_k y_k^\top, \overline{Z} \right\rangle_F = \langle \overline{Z}, \overline{Z} \rangle_F = \|\overline{Z}\|_F^2 = \|X^\top Y\|_F^2. \tag{23}$$

Finally, applying some minor substitutions we get equation equation 9 and finish the proof.

$\square$

### A.2    PROOF OF THE LEMMA 2.2

**Lemma 2.2** (Apriori variance of RMM). *Let $X \in \mathbb{R}^{B \times N}$ and $Y \in \mathbb{R}^{B \times M}$, then the variance of a randomized matrix multiplication through a matrix $S \in \mathbb{R}^{B \times B_{\text{proj}}}$ with i.i.d. elements following the normal distribution $\mathcal{N}(0, B_{\text{proj}}^{-0.5})$ defined as*

$$D^2(X, Y) = \mathbb{E}_S \|X^\top S S^\top Y - X^\top Y\|_F^2 \tag{10}$$

*can be evaluated as follows*

$$D_{\text{RMM}}^2(X, Y) = \frac{\|X\|_F^2 \|Y\|_F^2 - \|X^\top Y\|_F^2}{B_{\text{proj}}}. \tag{11}$$

*Proof.* So, we are interested in the deviation of the randomly sampled observations $X^\top SS^\top Y$:

$$D^2(X,Y) = \mathbb{E}_S \|X^\top SS^\top Y - X^\top Y\|_F^2. \tag{24}$$

The square Frobenius norm can actually be rewritten with a help of a trace of a matrix:

$$D^2(X,Y) = \mathbb{E}_S \left[\operatorname{tr}\left((X^\top SS^\top Y - X^\top Y)(X^\top SS^\top Y - X^\top Y)^\top\right)\right]. \tag{25}$$

Due to linearity of the trace we obtain

$$D^2(X,Y) = \mathbb{E}_S \left[\operatorname{tr}\left(X^\top SS^\top YY^\top SS^\top X\right)\right] - \operatorname{tr}\left(X^\top YY^\top X\right). \tag{26}$$

The trace is invariant under a certain shift of multipliers:

$$\operatorname{tr}\left(X^\top SS^\top YY^\top SS^\top X\right) = \operatorname{tr}\left(S^\top XX^\top SS^\top YY^\top S\right). \tag{27}$$

Let assume some positive values $a$ and $b$ such that $4ab = 1$ and introduce the following symmetric matrices:

$$A = aXX^\top + bYY^\top, \quad B = aXX^\top - bYY^\top. \tag{28}$$

Hence, we produce the following substitution:

$$\operatorname{tr}\left(S^\top XX^\top SS^\top YY^\top S\right) = \operatorname{tr}\left(S^\top ASS^\top A^\top S\right) - \operatorname{tr}\left(S^\top BSS^\top B^\top S\right) = \|S^\top AS\|_F^2 - \|S^\top BS\|_F^2. \tag{29}$$

Similar steps can be applied to the other trace in equation equation 26:

$$\operatorname{tr}\left(X^\top YY^\top X\right) = \operatorname{tr}\left(XX^\top YY^\top\right) = \operatorname{tr}\left(AA^\top\right) - \operatorname{tr}\left(BB^\top\right) = \|A\|_F^2 - \|B\|_F^2. \tag{30}$$

It is now possible to simplify the square deviation as

$$D^2(X,Y) = \left(\mathbb{E}_S \|S^T AS\|_F^2 - \|A\|_F^2\right) - \left(\mathbb{E}_S \|S^T BS\|_F^2 - \|B\|_F^2\right). \tag{31}$$

Since the matrix $A$ is symmetric, it is diagonalizeable:

$$A = Q^\top \Lambda Q, \tag{32}$$

with an orthogonal matrix $Q \in \mathbb{R}^{B \times B}$ and a diagonal matrix $\Lambda \in \mathbb{R}^{B_{proj} \times B_{proj}}$. While $S$ consists of i.i.d. random values following the normal distribution $\mathcal{N}(0, B_{proj}^{-0.5})$, the same is true for a matrix $C = QS \in \mathbb{R}^{B \times B_{proj}}$, and therefore

$$\mathbb{E}_S \|S^T AS\|_F^2 = \mathbb{E}_C \|C^T \Lambda C\|_F^2. \tag{33}$$

Let us estimate the latter value:

$$\mathbb{E}_C \|C^T \Lambda C\|_F^2 = \sum_{i=1}^{B_{proj}} \sum_{j=1}^{B_{proj}} \mathbb{E}_C \left(\sum_{l=1}^{B} \lambda_l C_{li} C_{lj}\right)^2. \tag{34}$$

In the case $i \neq j$:

$$\mathbb{E}_C \left(\sum_{l=1}^{B} \lambda_l C_{li} C_{lj}\right)^2 = \sum_{l=1}^{B} \sum_{p=1}^{B} \lambda_l \lambda_p \mathbb{E}_C (C_{li} C_{lj} C_{pi} C_{pj}) = \sum_{l=1}^{B} \lambda_l^2 \mathbb{E}(C_{li}^2) \mathbb{E}(C_{lj}^2) = B_{proj}^{-2} \operatorname{tr}(A^2). \tag{35}$$

In the case $i = j$:

$$\mathbb{E}_C \left(\sum_{l=1}^{B} \lambda_l C_{li} C_{lj}\right)^2 = \sum_{l=1}^{B} \sum_{p=1}^{B} \lambda_l \lambda_p \mathbb{E}_C (C_{li}^2 C_{pi}^2) = \left(\sum_{l=1}^{B} \lambda_l \mathbb{E}(C_{li}^2)\right)^2 = B_{proj}^{-2} (\operatorname{tr}(A))^2. \tag{36}$$

Accumulating all possible values $i = 1, \ldots, B_{proj}$ and $j = 1, \ldots, B_{proj}$ we obtain the following result:

$$\mathbb{E}_S \|S^T AS\|_F^2 = \left(1 - B_{proj}^{-1}\right) \operatorname{tr}(A^2) + B_{proj}^{-1} (\operatorname{tr}(A))^2 \tag{37}$$

Subtracting the Frobenius norm of $A$ we get

$$\mathbb{E}_S \|S^T AS\|_F^2 - \|A\|_F^2 = B_{proj}^{-1} \left((\operatorname{tr}(A))^2 - \operatorname{tr}(A^2)\right) \tag{38}$$

Coming back to the square of the deviation we obtain:

$$D^2(X,Y) = B_{proj}^{-1} \left[\operatorname{tr}(A - B)\operatorname{tr}(A + B) - \operatorname{tr}((A - B)(A + B))\right]. \tag{39}$$

The first summand is the following:

$$\operatorname{tr}(A - B)\operatorname{tr}(A + B) = 4ab\operatorname{tr}(XX^\top)\operatorname{tr}(YY^\top) = \|X\|_F^2 \|Y\|_F^2. \tag{40}$$

The second summand is the following:

$$\operatorname{tr}((A - B)(A + B)) = 4ab\operatorname{tr}(XX^\top YY^\top) = \operatorname{tr}(X^\top YY^\top X) = \|X^\top Y\|_F^2. \tag{41}$$

Substituting equations equation 40 and equation 41 into equation equation 39 we finish the proof.

$$\square$$

### A.3 Proof of the Theorem 2.3

**Theorem 2.3** (Upper bound of variance). *In the conditions of Lemma 2.1 and Lemma 2.2 the in-sample variance $D_{\mathrm{SGD}}$ and the variance $D_{\mathrm{RMM}}$ induced by a randomized subsampling are tied with the following inequality*

$$\frac{B_{\mathrm{proj}}}{B-1} \frac{D^2_{\mathrm{RMM}}(X,Y)}{D^2_{\mathrm{SGD}}(X,Y)} \leq \frac{\alpha+1}{\alpha}, \tag{12}$$

*where*

$$\alpha = \frac{\|X^\top Y\|^2_F}{\|X\|^2_F \|Y\|^2_F} \in [0,1]. \tag{13}$$

*Proof.* Let us introduce the following correlation ratio:

$$\alpha = \frac{\|X^\top Y\|^2_F}{\|X\|^2_F \|Y\|^2_F} \in [0,1]. \tag{42}$$

Now let us evaluate the following difference:

$$B_{proj} D^2_{\mathrm{RMM}}(X,Y) - (B-1)\frac{\alpha+1}{\alpha} D^2_{\mathrm{SGD}}(X,Y) = \|X\|^2_F \|Y\|^2_F - \|X^\top Y\|^2_F - B\frac{\alpha+1}{\alpha} \sum_{i=1}^{B} \|x_i\|^2 \|y_i\|^2 \tag{43}$$

$$+ \frac{\alpha+1}{\alpha} \|X^\top Y\|^2_F. \tag{44}$$

It is clearly reduced to the following statement:

$$B_{proj} D^2_{\mathrm{RMM}}(X,Y) - (B-1)\frac{\alpha+1}{\alpha} D^2_{\mathrm{SGD}}(X,Y) = -B\frac{\alpha+1}{\alpha} \sum_{i=1}^{B} \|x_i\|^2 \|y_i\|^2 \leq 0. \tag{45}$$

So we finish proving the inequality.

$\square$

## B   Details of Experiments

In this section we presents more detailed experimentation results. RoBERTa model was fine-tuned with PyTorch Paszke et al. (2019) and HuggingFace's Transformers Wolf et al. (2020). All hyperparameters and experimental settings in fine-tuning on GLUE are taked from Fairseq Ott et al. (2019). The only difference is that we use batch size 16 instead of 32 for QNLI task since peak memory usage exceeds 16 GiB in training time. We assume Gaussian randomized matmul whereever the opposite is not indicated.

### B.1 Variance Estimation

We train RoBERTa model on GLUE benchmark. We use a dense layer in output of transformer block #7 for all experiments related to empirical variance estimation. Auxiliary values tracked in fine-tuning on GLUE are shown on Figure 5.

### B.2 Learning Curves

See Figure 6 for loss curves on training set and evaluation set.

### B.3 RMM on Graph Neural Networks in Comparison with EXACT

In this section we compare RMM on non-textual domain (graph neural networks) versus EXACT (Liu et al., 2021) according to experimental protocols from (Liu et al., 2021) (see Table 5 and Figure 7).

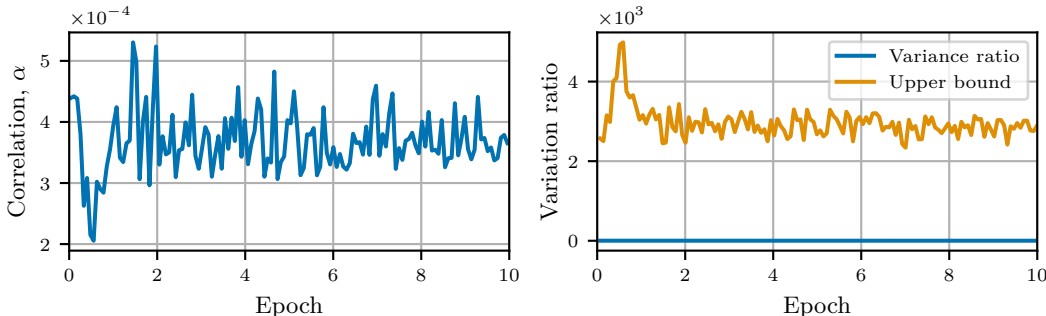

Figure 5: Evolution of correlation coefficient $\alpha$ and variances equation 9 and equation 11 during fine-tuning on CoLA for batch size $B = 64$ and compression rate $\rho = 0.5$.

Table 4: Target metric averaged on 3 runs on downstream tasks of GLUE dataset. The top line in the table means that no compression techique was used.

| RATE | CoLA | MRPC | QNLI | RTE |
|---|---|---|---|---|
| — | $60.51 \pm 1.31$ | $89.30 \pm 0.93$ | $92.60 \pm 0.13$ | $78.52 \pm 2.29$ |
| 90 | $59.75 \pm 1.14$ | $88.64 \pm 0.37$ | $92.75 \pm 0.04$ | $77.50 \pm 1.04$ |
| 50 | $59.45 \pm 1.23$ | $88.73 \pm 0.49$ | $92.56 \pm 0.22$ | $77.18 \pm 1.06$ |
| 20 | $57.46 \pm 1.21$ | $87.99 \pm 0.49$ | $92.62 \pm 0.17$ | $76.26 \pm 1.99$ |
| 10 | $57.53 \pm 1.17$ | $88.30 \pm 0.54$ | $92.55 \pm 0.13$ | $75.45 \pm 1.08$ |

| RATE | SST2 | STS-B | WNLI |
|---|---|---|---|
| — | $94.09 \pm 0.11$ | $90.37 \pm 0.18$ | $56.34 \pm 0.00$ |
| 90 | $94.72 \pm 0.28$ | $90.39 \pm 0.18$ | $54.93 \pm 3.15$ |
| 50 | $94.61 \pm 0.57$ | $90.32 \pm 0.23$ | $56.34 \pm 0.00$ |
| 20 | $94.43 \pm 0.59$ | $90.06 \pm 0.13$ | $56.34 \pm 0.00$ |
| 10 | $94.27 \pm 0.50$ | $89.90 \pm 0.19$ | $47.89 \pm 7.32$ |

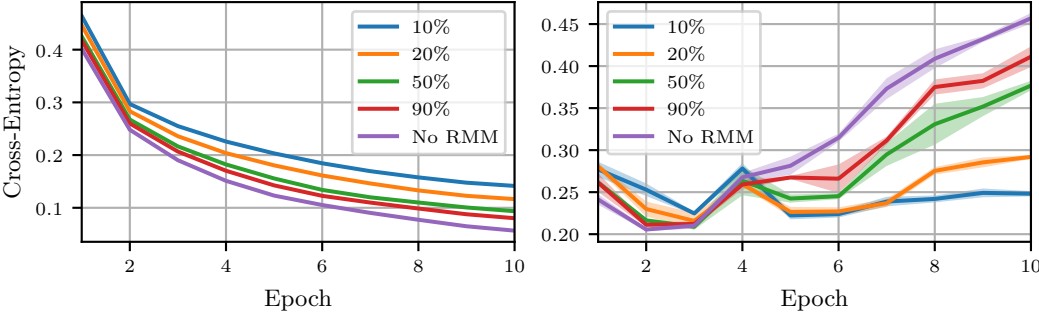

Figure 6: Train and test loss averaged accross 3 runs for ROBERTA fine-tuned on QNLI task from GLUE benchmark.

## B.4 MEMORY USAGE

For more extensive exploration of memory usage for various GLUE tasks see Figure 8.

Table 5: Comparison of EXACT and RMM (ours) on graph neural network GCN2 on OBGN-ARXIV dataset. Both methods are applied with the same compression rate $\rho = 0.1$. Each experiment is repeated 3 times.

| METHOD | TEST ACCURACY | RATIO |
|---|---|---|
| BASELINE | $72.87 \pm 0.68$ | $0.0\%$ |
| EXACT | $72.61 \pm 0.27$ | $-0.36\%$ |
| RMM (GAUSS) | $70.99 \pm 0.33$ | $-2.58\%$ |
| RMM (RADEMACHER) | $71.53 \pm 0.13$ | $-1.84\%$ |

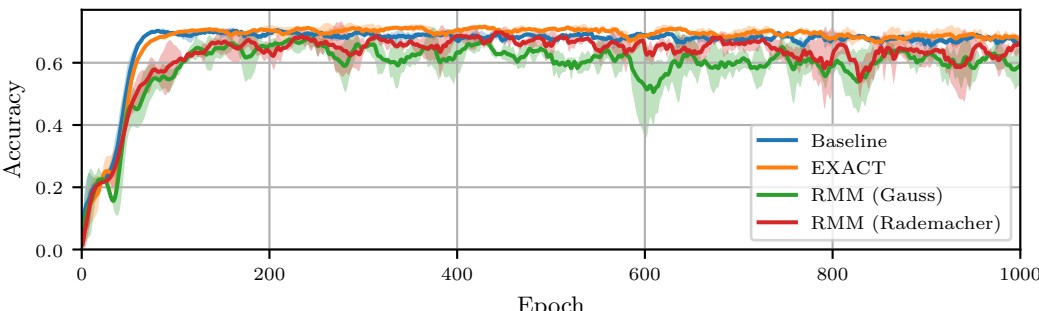

Figure 7: Test accuracy averaged accross 3 runs for GCN2 graph neural network trained on OBGN-ARXIV dataset. Memory saving method EXACT and RMM (ours) are applied with the same compression rate $\rho = 0.1$.

## C  METHOD DETAILS

### C.1  COMPLEXITY

#### C.1.1  COMPUTATIONAL COMPLEXITY

Let $B$ denote the batch dimension and $N_{\text{in}}$ and $N_{\text{out}}$ be the input and the output sizes of a linear layer respectively. Also, let the compression rate $\rho \in (0, 1]$ and the compressed batch dimension $B_{\text{proj}} = \rho B$.

According to Alg. 1 the forward pass of a linear layer requires $O(BN_{\text{in}}N_{\text{out}})$ operations to compute the output $\hat{X}$ and $O(BB_{\text{proj}}N_{\text{in}})$ operations to obtain the compressed input $X_{\text{proj}}$ for the backward pass.

Arithmetical complexity of the baseline backward pass, which is based on the non-compressed input $X$, is $O(BN_{\text{in}}N_{\text{out}})$ floating point operations. On the other hand, our approach for the backward pass requires multiplication of the output gradients by a rematerialized random matrix $S$ and estimation of the gradients with respect to weights resulting in $O(BB_{\text{proj}}N_{\text{out}} + B_{\text{proj}}N_{\text{in}}N_{\text{out}})$ operations. Total asymptotic complexity of a single forward-backward cycle is $O(BN_{\text{in}}N_{\text{out}})$ for the baseline implementation and $O(B_{\text{proj}}N_{\text{out}}(B + N_{\text{in}}))$ for our approach.

Assume that $N = N_{\text{in}} \sim N_{\text{out}}$ then overall complexities become $O(BN^2)$ and $O(\rho BN(B + N))$, respectively. In real world scenarios of large Transformer models with $N \ll B$, we conclude to $O(B^2 N)$ operations where compression rate is merged into a constant multiplier. Randomized matmul modification of a linear layer has worse asymptotic in terms of the batch size but choosing small enough compression rate $\rho$ reduces computational time significantly and makes our approach practically appealing.

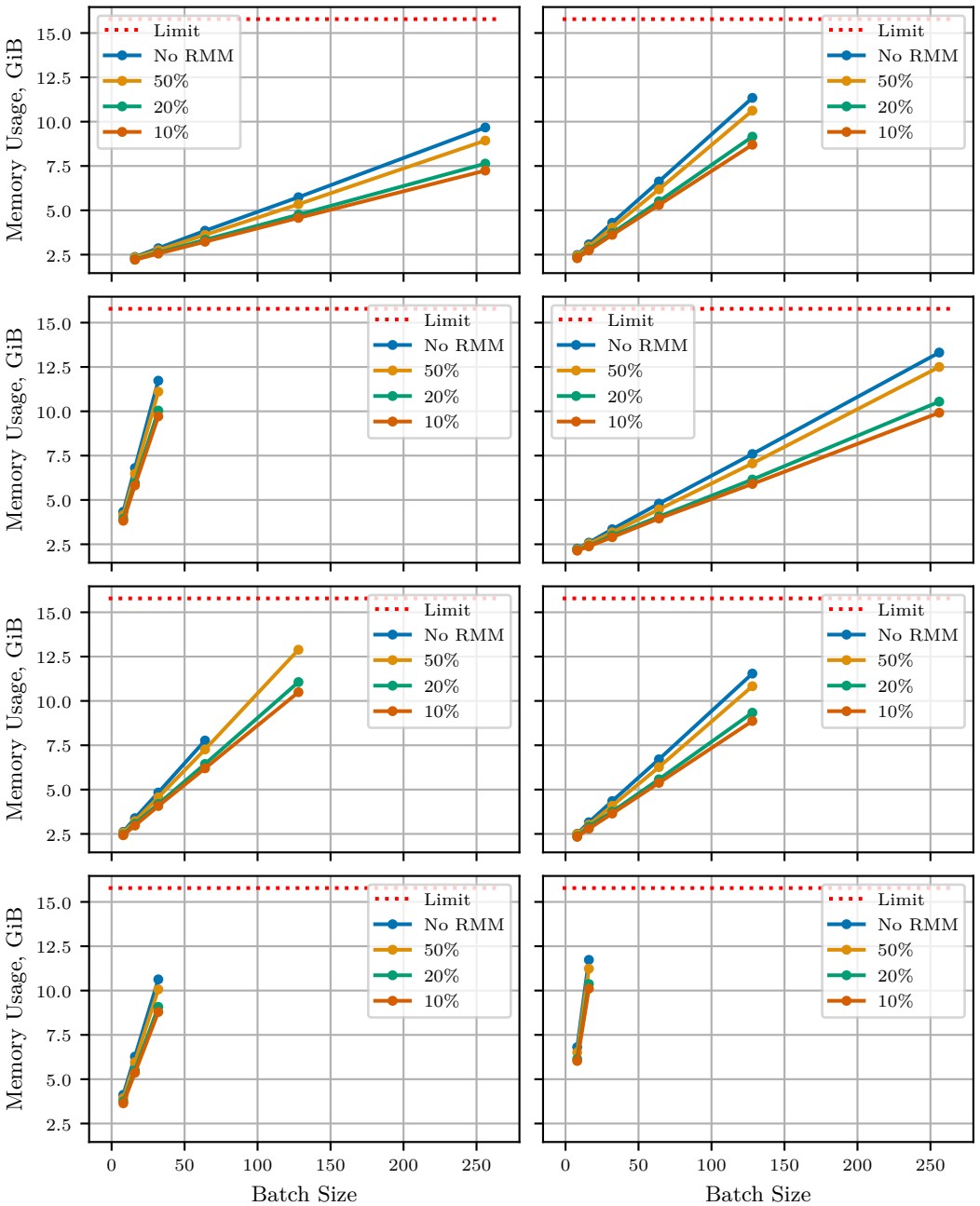

Figure 8: Memory usage during training on GLUE tasks during for epoch with randomized Gaussian matmul (from left to right and from top to bottom CoLA, MRPC, QQP, SST2, STSB, WNLI, RTE, and QNLI).