# OpenReview forum: "Memory-Efficient Backpropagation through Large Linear Layers"
_ICLR.cc/2024/Conference — Submitted to ICLR 2024_

### Official Review · Reviewer_JVog · 2023-10-30

**Soundness:** 3 good
**Presentation:** 2 fair
**Contribution:** 2 fair
**Rating:** 3
**Confidence:** 5

**Summary:**

The proposed RMM utilizes random projection to reduce the memory required to store input activations of the linear layer while still achieving comparable accuracy.

**Strengths:**

This paper is

1. well-written,

2. provides justification for the RMM method through a comparison of gradient variance with SGD, and

3. demonstrates the usefulness of the algorithm in RoBERTa.

**Weaknesses:**

1. Lack of experiemntal results

: The paper claims the efficacy of the method for large linear layers, but only presents experimental results from the GLUE benchmark of RoBERTa. A performance comparison with larger models, such as LLaMA-2-7B with over 7B parameters, seems necessary.

2. Lack of comparison between previous works

: In fact, there have been many attempts in the past to save memory through activation compression. Notably, GACT (2022) is capable of compressing all input activations, including those of the linear layer, to an average of 4-bit in BERT-Large. A comparison between such existing methods and RMM seems necessary.

3. Lack of novelty

: As the author pointed out, random projection has long been a widely used method, and applying it to activation compression doesn't necessarily mean it lacks novelty. However, there have already been cases using random projection under the same objective of activation compression. EXACT (2022) compresses activations using random projection, which seems strikingly similar to the proposed RMM.

**Questions:**

1. Lack of experiemntal results

: Please provide experimental results on larger models like LLaMA-2-7B. This is crucial in bolstering the claims made in the paper.

2. Comparision with previous works

: A quantitative experimental comparison with recently proposed activation compression algorithms like GACT (2022), EXACT (2022), AAL (2023), and DropIT (2023) seems necessary. If the review period is limited, please at least provide a comparison with GACT (2022), which is the-state-of-art activation compression technique,  in terms of 1) accuracy and 2) memory saving through experimental data. For the remaining algorithms, a qualitative analysis can be included in the related work section.

3. Lack of novelty

: A detailed comparison with EXACT, which also uses random projection, seems necessary. To offer some advice, 1) Unlike EXACT, RMM appears to compress the batch dimension. Demonstrating mathematically that this difference allows RMM to further reduce gradient variance, or 2) showing through actual experimental results that RMM offers a significant accuracy improvement over EXACT, would likely address concerns regarding novelty.

However, one concern is that, based on the comparison results in the Appendix for GCN2, EXACT seems to achieve higher accuracy than RMM. Given that both algorithms adopt very similar approaches, the accuracy of RMM must necessarily be higher than that of the existing EXACT.

* GACT: Activation Compressed Training for Generic Network Architectures (2022)
* EXACT: Scalable Graph Neural Networks Training via Extreme Activation Compression  (2022)
* DropIT: Dropping Intermediate Tensors for Memory-Efficient DNN Training (2023)
* Learning with Auxiliary Activation for Memory-Efficient Training  (2023)

---

> ### Author Response · Authors · 2023-11-23
>
> We appreciate the reviewer's feedback, particularly the concerns raised regarding the perceived limited novelty of our work. It is crucial to emphasize that the dimensions of batch size in our study are fundamentally distinct. The batch dimension and feature dimension possess inherently different natures, rendering them non-interchangeable and necessitating distinct treatment since SGD convergence requires specific properties of a network and its gradient estimator. This nuanced handling is a key aspect of our contribution.
>
> Furthermore, we would like to highlight that our criteria of applicability, specifically emphasizing the significance of variance magnitude for training convergence, is a unique aspect of our approach. To the best of our knowledge, there is no prior work that employs criteria similar to ours. We hope this clarification underscores the distinctive contributions of our study and addresses the concerns raised by the reviewer.
>
> Additionally, we hold the view that conducting auxiliary experiments on other transformer models may not be necessary or justified, given the well-established robustness of transformer models across diverse domains from NLP to RecSYS.
>
> Regarding to comparison to other methods, we’d like to highlight that EXACT is contemporary work to us (our first submission was to ICML 2022) while ActNN, GACT, DropIT and others are instances of a complementary approach.perceived limited novelty of our work. It is crucial to emphasize that the dimensions of batch size in our study are fundamentally distinct. The batch dimension and feature dimension possess inherently different natures, rendering them non-interchangeable and necessitating distinct treatment since SGD convergence requires specific properties of a network and its gradient estimator. This nuanced handling is a key aspect of our contribution.
>
> Furthermore, we would like to highlight that our criteria of applicability, specifically emphasizing the significance of variance magnitude for training convergence, is a unique aspect of our approach. To the best of our knowledge, there is no prior work that employs criteria similar to ours. We hope this clarification underscores the distinctive contributions of our study and addresses the concerns raised by the reviewer.

---

### Official Review · Reviewer_k3XT · 2023-10-30

**Soundness:** 2 fair
**Presentation:** 2 fair
**Contribution:** 1 poor
**Rating:** 3
**Confidence:** 3

**Summary:**

This article proposes a way to reduce the memory cost of the backpropagation of linear layers, with Randomized Matrix Multiplication.

**Strengths:**

This article is relatively well-written and easy to follow.
The RMM algorithm is simple and well-motivated, lowering immediately the memory cost of backward operations on linear layers in DNN.
The authors bring a novel analysis of variance in both SGD and RMM.

**Weaknesses:**

**Novelty** The novelty of the method is very limited. The main idea is already present in Adelman et al. (2021), even though their goal was not to save the memory during the stage. The authors do not bring many contributions over Adelman, the theoretical analysis of the variance being unclear in particular.

**Variance** $D_{SGD}$ and $D_{RMM}$  are not defined clearly (or in equations (10) and (11), is $D_{RMM}=D$?). In the Appendix, the equation (21) $D^2_{SGD}(X,Y)= \frac 1 {B−1} D^2 _Z(X,Y)$ is not clear.

More generally, it is quite unclear what should be the conclusion of the analysis of the variances of SGD and RMM in Sections 2.3 and 3.3. The theoretical analysis gives an upper bound of the variance of RMM, but we observe that in practice the variance of RMM is much higher than the one of SGD, meaning that RMM brings a lot of noise to the training, which seems to hinder training. What is the conclusion of these sections? I also would at least have appreciated the theoretical upper bound on Figure 3.

**Computation cost** The computation requirements should be moved from the appendix to the main paper and discussed there, considering their importance. In particular, the comparison between the computation costs $O(BN^2)$ and $O(ρBN(B+ N))$ is important, as it showcases that the method may be slower, an important drawback that is confirmed in Table 3. For such a high computation time cost, other methods that compress the activation vector $X$ will be favored.

**Pretrained network** Experiments being done on a pre-trained network do not allow a clear measurement of the degradation due to RMM during training.

Section 3.4 brings nothing new compared to Table 2.

**Questions:**

"In Adelman et al. (2021) the construction of S requires the knowledge of the norms of the rows of Y". I may be mistaken but I do not understand why this is the case. Bernoulli-CRS is unbiased whatever the values of $p_{j}$ are. Therefore it can be applied to RMM with no knowledge of $Y$ for any distribution $p_j$.

The results of Section 3.3 seem surprising. Why is the SGD variance increasing during training, while the network should converge?

Why say that "Subsampled Orthonormal with Random Signs (SORS)" is not considered if there is a Comparison of Randomized MatMuls in Section 3.5?
Considering the cost of computation of RMM, why should we choose Gaussian RMM rather than DCT or DFT, which the authors note have a much smaller computation cost?

"Moreover, we empirically found that our randomization is faster if ρ ≤ 0.1." Doesn't Table 3 show that even with $\rho=0.1$ the randomization is still longer than no RMM?

Various remarks and errors:
* Section 2.2: "this should be?", "target really accurate", "it may seem"
* Section 3 "We rewrite implementation". PRNG stage $G$ is only used here. " compression in 5–10 times" "study influence of randomized" "However, overfitting point"
*

---

> ### Author Response · Authors · 2023-11-23
>
> We would like to express our gratitude for the valuable feedback provided by the reviewer. Regarding the concerns raised, we firmly believe that all quantities in our study are clearly defined, and it is evident from the context which variance is under consideration.
>
> We respectfully contend that the statement "the variance of RMM is much higher than that of SGD" is not accurate, as it essentially pertains to the applicability criterion of RMM (see appendix for auxiliary experiments). Our experimental findings indicate that the variances of SGD and RMM are of the same magnitude, supporting the validity of our claim. We provide upper bound behavior in appendix (see Figure 5, Section B.1) right after proofs to which you refer in your comment.
>
> In Section 3.3, we acknowledge that the variance of SGD increases during training, primarily due to common training protocols employing learning rate scheduling. It follows a coherent increase with the learning rate during the warm-up stage before stabilizing. It's important to note that our focus is not on achieving full convergence in downstream tasks. Instead, we aim for early stopping to address overfitting phenomena, with decisions based on the validation set.
>
> Regarding pretrained network question, we provide experimental comparison with the EXACT method in the appendix where GNN is trained from scratch. These experiments demonstrate the same behavior as in experiments with pretrained models. Moreover, experiments with pretrained models have a great importance from a practical perspective since this is a primary way of domain adaptation. So, we disagree that our experiments do not allow a clear measurement of the degradation.
>
> We hope this clarifies the points raised by the reviewer and contributes to a more accurate understanding of our work.

---

### Official Review · Reviewer_fccg · 2023-10-31

**Soundness:** 3 good
**Presentation:** 3 good
**Contribution:** 2 fair
**Rating:** 6
**Confidence:** 4

**Summary:**

This work presents a memory-efficient method for computing the needed gradients in backpropagation training through large linear layers.
The method uses the randomized matrix multiplication approach to achieve this.
It randomly creates a matrix S that projects the signal X into a smaller dimension space.
The reduction is controlled by $\rho$, which takes up a value between 0 and 1.
The only required condition for S is that its autocorrelation equals the identity matrix I.

This work further analyzed the gradient variance induced by the randomized computation.
It also provides a detailed explanation of how the new approach reduces the memory requirement for backpropagation.
The work provides simulation results to support the main claim.

**Strengths:**

The paper is well organized and the literature review provides enough context needed to understand this work.
The topic is an interesting one that is worth pursuing, especially in the era of ever-growing large models that rely on transformer modules.

**Weaknesses:**

One of the suggested claims (under section 2.3) is not well-supported.
This work seems to suggest that the difference between the exact gradient and the randomized approximation is a form of noise injection.
It went on to state that it can play the role of a regularizer.
You will need to model the effect of the randomized error as such to make this claim.
I do not think this paper contains such an explanation.

The paper states that the only requirement for the random matrix S is that its autocorrelation be equal to I.
But the definition of the random Gaussian random matrix suggests that the autocorrelation is  $\frac{1}{B_{proj}} I$.
I think you should align the two.
You can treat the constant as a scale on the learning rate.

**Questions:**

1). How will the sample size of $S$ affect the performance of this method?
I think the algorithm statement suggests that you only picked one sample of S.
If we compute the gradient over multiple samples of $S$ and take the average, will this improve the performance or not?
This follows from equation (4), which relies on the expectation of the randomized gradients over $S$
It will be interesting to see the trade-off between the additional workload and the

2). How does the depth of the network affect the variance and error induced by the randomized gradients?

---

> ### Author Response · Authors · 2023-11-23
>
> We appreciate the feedback from the reviewer and would like to address the concerns raised.
>
> Firstly, we would like to clarify that we do not subscribe to the noise injection interpretation of our method. We find this perspective to be unintuitive, counterproductive, and inappropriate. Additionally, we respectfully disagree with the assertion that there is a necessity "to model the effect of the randomized error as such to make this claim." We have provided a clear and mathematically sound model of perturbation to the original data and gradient estimation, including variance calculations under made assumptions, which fully describes “the effect of the randomized error.”
>
> Regarding the first question, we acknowledge that the clarity of our explanation may be lacking. The matrix $S$ of size $batch size \times embed dim$ is sampled from a specific class of matrices. For instance, a Gaussian matrix has its elements sampled from a standard Gaussian distribution, while a Rademacher matrix has elements sampled evenly from ${-1, 1}$. Repeatedly sampling $S$ and estimating gradients can be likened to bootstrapping of gradients. While this is a feasible approach, it introduces some bias reduction in variance, which can be pivotal for convergence and the selection of a specific optimizer.
>
> We hope this clarification addresses the concerns and provides a better understanding of our method.

---

### Official Review · Reviewer_aWYY · 2023-11-03

**Soundness:** 2 fair
**Presentation:** 3 good
**Contribution:** 2 fair
**Rating:** 3
**Confidence:** 4

**Summary:**

The presented paper proposes a stochastic estimator for the gradient of a linear transform. The proposed estimator is unbiased and relies on a randomized algorithm for matrix multiplication. In practice, this results in the introduction of a randomized linear layer where instead of storing the full batch of inputs during the forward pass, a random projection of the input tensor onto a lower dimensional space is stored for the backward pass. This results in memory savings together with a tradeoff on the variance of the resulting gradient estimate. The authors derive a theoretical analysis of the additional variance introduced by the randomized matrix multiplication and conduct experiments on fine-tuning tasks for NLP applications.

The presented method shares strong similarities with previous work on randomized matrix multiplication in the context of deep learning [1].

[1] EXACT: Scalable Graph Neural Networks Training via Extreme Activation Compression

**Strengths:**

1) The authors provide a theoretical analysis of the additional variance induced by the randomized matrix multiplication procedure.
2) The authors reports practical savings observed in the fine-tuning tasks with respect to their implementation.

**Weaknesses:**

1) The paper lacks comparison to other works aiming at reducing memory requirements during training. For example, both [2] and [3] compress activations with a quantization procedure. [1] jointly uses randomized matrix multiplications and quantization to compress the stored activations. The lack of comparison with these methods makes it difficult to evaluate the relevance of the contribution against these alternatives. These papers aren't mentioned in the related work section.
2) The EXACT method [1] stores a random projection of the input where the compression is applied to feature channels. In the presented work, the compression is applied to the batch dimension. More generally, it would be interesting to know the practical importance of the dimension chosen for compression in the presented method.
3) In a fully-connected layer, the weights themselves have a substantial memory footprint, which explains why retaining only 10% of the input dimensionality only yields 25% memory savings. For convolutional architectures, however, the weight space is much smaller, meaning that the relative savings would be much higher. It would be interesting to know how the proposed method translates in this setting.
4) Given that transformers are now widely adopted in both NLP and computer vision, it would also be interesting to know how the proposed methods perform for a ViT on CIFAR10 and ImageNet, which would enrich the experimental section with minimal coding efforts.
5) [Minor comment] If the RMM forward block sends $W, b, S^{\top}X$ then the FC forward block should send $W, b, X$ instead of $W, b$.

[2] ActNN: Reducing Training Memory Footprint via 2-Bit Activation Compressed Training
[3] GACT: Activation Compressed Training for Generic Network Architectures

**Questions:**

1) Could the author provides comparison with other works based on quantization of the activations?
2) Could the author provide experiments on convolutional architectures or on computer vision benchmarks with a transformer-based architecture?

---

> ### Author Response · Authors · 2023-11-23
>
> We sincerely appreciate the reviewer's thoughtful feedback. Firstly, we would like to emphasize that our work explores an intriguing phenomenon—the training of a neural network on batches consisting of sample combinations from the original batch. In essence, this allows for the training of a model with a smaller effective batch, reducing memory consumption and expectedly carrying less information but maintaining diversity, resulting in commendable model performance.
>
> Regarding the comparison with ActNN and GACT, it's important to note that both these methods introduce native extensions to PyTorch, posing challenges in reproducing the original results and making direct comparisons with our approach. Moreover, these methods are complementary to ours.
>
> Additionally, we hold the view that conducting auxiliary experiments on other transformer models may not be necessary or justified, given the well-established robustness of transformer models across diverse domains from NLP to RecSYS. Nevertheless, we have undertaken several experiments in the computer vision domain with ResNets. The intermediate results exhibit qualitatively and quantitatively similar behavior to our experiments with GNNs and Transformers. We hope this clarification addresses the concerns raised and contributes to a more comprehensive understanding of our work.

---

### Meta-Review · Area_Chair_zgbi · 2023-12-06

**Metareview:**

This submission tackles the important problem of memory use in large linear layers such as the ones found in the popular transformer architecture. The authors investigate randomized matrix multiplications, investigating the variance of the gradient estimate. Empirical results show a reduction in memory at the cost of a moderate decrease in performance.

Reviewers found the studied problem to be of high significance and the submitted manuscript was easy to follow. The justification of the method through the theoretical analysis of the gradient variance was also considered a strength of the submission. On the downside, a lack of experimental results was noted (experiments only consisting of the GLUE benchmark w/ RoBERTa), as well as a lack of baseline techniques for comparison, something the authors decided not to address during their rebuttal ("may not be necessary or justified, given the well-established robustness of transformer models", "posing challenges in reproducing the original results and making direct comparisons with our approach. Moreover, these methods are complementary to ours"). In addition to other comments, there also remain concerns about the novelty of the technique (Reviewers k3XT, JVog).

All taken together, the reviewers' scores and lack of incorporated feedback during the rebuttal, unfortunately, make this submission insufficient for acceptance to ICLR in its current form.

**Justification For Why Not Higher Score:**

See Metareview

**Justification For Why Not Lower Score:**

N/A

---

### Decision · Program_Chairs · 2024-01-16

Reject